# Evaluation of masticatory muscles in temporomandibular joint disorder patients using quantitative MRI fat fraction analysis—Could it be a biomarker?

**Kug Jin Jeon, Yoon Joo Choi, Chena Lee, Hak-Sun Kim, Sang-Sun Han** *

Department of Oral and Maxillofacial Radiology, Yonsei University College of Dentistry, Seoul, Republic of Korea

* sshan@yuhs.ac

**Data Availability Statement:** All relevant data are within the manuscript.

**Funding:** Yonsei University College of Dentistry (6-2023-0005) The funders had no role in study

## Abstract

Temporomandibular joint disorders (TMDs) are closely related to the masticatory muscles, but objective and quantitative methods to evaluate muscle are lacking. IDEAL-IQ, a type of chemical shift-encoded magnetic resonance imaging (CSE-MRI), can quantify the fat fraction (FF). The purpose of this study was to develop an MR IDEAL-IQ-based method for quantitative muscle diagnosis in TMD patients. A total of 65 patients who underwent 3 T MRI scans, including CSE-MRI sequences, were retrospectively included. MRI diagnoses and clinical data were reviewed. There were 19 patients in the normal group and 46 patients in the TMD group with unilateral disc displacement. The TMD group was subdivided into those with and without clenching. The right and left FF values of the masseter, medial, and lateral pterygoid muscles were measured twice by two oral radiologists on CSE-MRI, and the average value was used. FF measurements using CSE-MRI showed excellent intra- and inter-observer agreement (ICC > 0.889 for both). There were no statistically significant differences between the right and left FF values in the masseter, medial pterygoid, and lateral pterygoid of the normal group ($p > 0.05$). A statistically significant difference was found in the TMD group without clenching, in which the masseter muscle had a statistically significantly lower FF value on the disc displacement side (3.94 ± 1.61) than on the normal side (4.52 ± 2.24) ($p < 0.05$). CSE-MRI, which can reproducibly quantify muscle FF values, is expected to be a biomarker for objective muscle evaluation in TMD patients. The masseter muscle is expected to be particularly useful compared to other masticatory muscles, but further research is needed.

## Introduction

The prevalence of temporomandibular joint disorders (TMDs) is gradually increasing and the age of onset is decreasing, with TMDs present in approximately 31% of adults and older adults and 11% of children and adolescents [1]. TMDs can involve disc disorders due to disc displacement, osteoarthritis with bone changes, and muscle abnormalities such as myalgia and

design, data collection and analysis, decision to publish, or preparation of the manuscript.

**Competing interests:** This work was supported by a grant from the National Research Foundation of Korea, funded by the Korean government (NRF-2022R1A2B5B01002517). The funders had no role in study design, data collection and analysis, decision to publish, or preparation of the manuscript.

myospasm. Temporomandibular joint (TMJ) and masticatory muscles are functionally closely related, so muscle abnormalities may occur along with disc displacement and osteoarthritis. TMDs are typically diagnosed based on magnetic resonance imaging (MRI) for disc abnormalities [2] and cone-beam computed tomography (CBCT) or computed tomography (CT) for degenerative bone changes [3–5]. Muscles are evaluated by palpation with 1 kg of pressure for 2 seconds, and if pain is present, the pressure is prolonged for 5 seconds to allow more time to elicit spreading or referred pain [6]. However, these methods are not objective and cannot be quantified, and assessing subtle changes in muscles can be difficult. To date, there is no quantitative evaluation method for muscles, although a few studies have measured muscle thickness using CT [7, 8] and MRI [9, 10]. Some research has assessed the thickness and elasticity of masseter muscles using ultrasound [11–13], but Blicharz et al. [14] reported in a literature review that a standardized method for evaluating masseter muscles is necessary. Furthermore, ultrasound signals cannot pass through bones and can only evaluate the superficially located masseter muscle and temporalis muscles, and it is unable to assess the deep-seated pterygoid muscles [15].

Chemical shift-encoded MRI (CSE-MRI) has emerged as a method capable of quantifying the proton density fat fraction (FF), including the iterative decomposition of water and fat with echo asymmetry and least-squares estimation (IDEAL-IQ) method [16]. CSE-MRI has been studied in various body parts [17–23], as well as diseases such as diabetes mellitus [24, 25] and malignant lesions [26]. In TMDs, only the FF values of the mandibular condyle have been studied [27], and Chen et al. [28] reported that changes in FF values in rabbits could be used as indicators of masticatory muscle dysfunction. Therefore, it was expected that FF values using IDEAL-IQ could quantitatively evaluate the masticatory muscle and be used as a biomarker for muscle diagnosis in TMDs. We hypothesized that in the masticatory muscles, the FF value on the disc displacement side would be different from the FF value of the normal side, but that in subjects with clenching, there would be no difference between both sides.

The present study was the first to investigate human masticatory muscles using FF values; specifically, we aimed to confirm the usefulness of FF values in evaluating the masticatory muscles of TMDs patients. First, we confirmed whether there was a difference in the FF values of the masticatory muscles on the right and left sides in normal subjects. Second, in patients with disc displacement, a type of TMDs, we compared and analyzed the FF values of the masticatory muscles on the normal side and the disc displacement side. Additionally, we compared the FF values with and without clenching, which affects muscles.

## Materials and methods

### Subjects

All 3 T MRI studies were reviewed from August 2022 to March 2023 at Yonsei University Dental Hospital, along with the corresponding electronic dental records. These data were accessed for research purposes from May 19 to September 20, 2023. An oral radiologist with 21 years of experience evaluated the disc position using MRI. A total of 65 patients were enrolled. Using the G*Power program, the sample size was calculated based on previous study (power = 0.9, α = 0.05) [23]. The study included only adults aged 18 years and older, and patients with bilateral disc displacement and any images with poor quality that could not be evaluated were excluded. Clinical symptoms were assessed according to the Diagnostic Criteria for Temporomandibular Disorders (DC/TMD) [6].

Individuals without TMD symptoms and with normal disc position were categorized into the normal group, while those with TMD symptoms and unilateral disc displacement were placed in the TMD group. The normal group included 19 participants (13 males, 6 females,

**Table 1. Clinical characteristics of the normal group and the TMD group with unilateral disc displacement.**

|  | Normal group | TMD group | |
|---|---|---|---|
|  | (n = 19) | Without clenching | With clenching |
|  |  | (n = 24) | (n = 22) |
| Male | 13 | 4 | 3 |
| Female | 6 | 20 | 19 |
| Age (years) | 37.21 ± 16.77 | 46.29 ± 16.67 | 36.05 ± 14.08 |

Age is presented as mean ± standard deviation.

aged 37.21 ± 16.77 years), and the TMD group consisted of 46 participants (7 males, 39 females, aged 41.39 ± 16.17 years). In the TMD group, the presence of clenching (or bruxism) that affected the masticatory muscles was diagnosed through self-report of clenching and a clinical examination to evaluate clenching signs, such as abnormal tooth wear and teeth impressions in the buccal region [29]. Twenty-four individuals had no clenching habit, while 22 exhibited a clenching habit (Table 1).

## CSE-MR imaging and fat fraction analysis of masticatory muscles

MRI scans of the masticatory muscles were obtained using a 3.0-T scanner (Pioneer; GE Healthcare Technology) with a 16-channel flex large coil. The 3D axial IDEAL-IQ sequence images were acquired with the following parameters: TR 11.0 ms, TE 4.0 ms, bandwidth 111.11 kHz, NEX 1.0, flip angle 4˚, field of view 24 × 24 cm, slice thickness 4.0 mm, and scan time 1 min. Images were acquired at multiple echo times. We acquired six different echoes ranging from 1.0 ms to 6.2 ms.

The FFs of the masseter muscles, medial pterygoid muscles, and lateral pterygoid muscles were analyzed using IDEAL-IQ in both the normal and TMD groups. One slice was selected for the region of interest (ROI) of each muscle. ROIs were drawn along each muscle on each axial image where the right and left muscles had the maximum area (Fig 1). Vessels and fat were excluded from the ROI. The temporal muscles were not clearly visible in many patients, so the FF of the temporal muscles was not measured. To assess reproducibility, two oral and

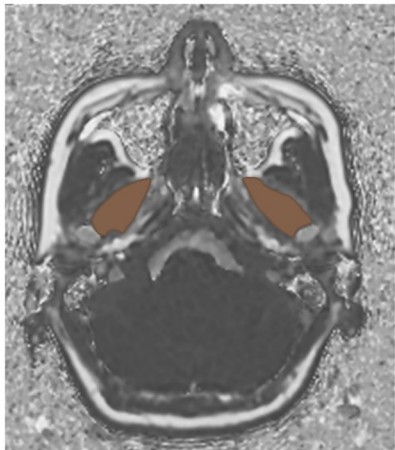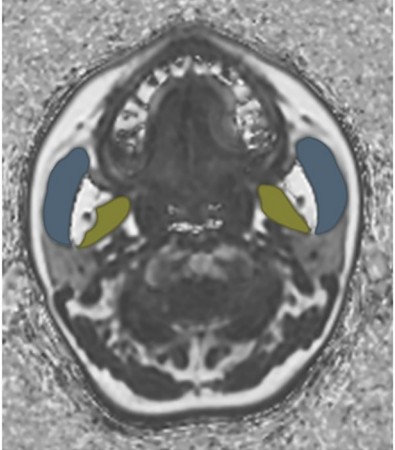

**Fig 1. Regions of interest in masticatory muscles.** The right and left lateral pterygoid muscles are orange, the medial pterygoid muscles are yellow, and the masseter muscles are blue.

**Table 2. Intra- and inter-observer agreement for two observers (95% CI).**

|  | Radiologist 1 | Radiologist 2 |
|---|---|---|
| **Intra-observer agreement** | 0.989 (0.984–0.992) | 0.889 (0.839–0.923) |
| **Inter-observer agreement** | 0.963 (0.946–0.974) | |

CI = confidence interval.

maxillofacial radiologists with 21 years and 13 years of experience, respectively, performed FF analysis twice at 1-month intervals in the normal group. In the TMD group, an oral and maxillofacial radiologist with 21 years of experience measured the FF twice at a monthly interval. The average value of the two measurements by the oral and maxillofacial radiologist with 21 years of experience was used for all statistical analyses.

## Statistical analysis

Intraclass correlation coefficients (ICCs) with 95% confidence intervals (CIs) were used to evaluate intra- and inter-observer agreement. The normality of data distribution was confirmed using the Kolmogorov-Smirnov test in all groups, including the normal group, TMD group, the subgroup without clenching, and the subgroup with clenching. In the normal group, the paired t-test was utilized to compare the FF values between the right and left sides. In the TMD group, the paired t-test was applied to compare the FF values between the normal side and the side with disc displacement. In the sub-analysis of the TMD group, the paired t-test was used to compare the FF values between the normal side and the side with disc displacement. Statistical analyses were conducted using GraphPad Prism version 9.5.1 (GraphPad Software). Statistical significance was defined as a $p$ value less than 0.05.

## Results

The two radiologists had intra-observer agreement values of 0.989 and 0.889, respectively. The inter-observer agreement was 0.963, indicating excellent agreement (Table 2).

No statistically significant differences were found between the right and left FF values in the masseter muscle, medial pterygoid muscle, and lateral pterygoid muscle of the normal group ($p > 0.05$) (Table 3).

In the TMD group, there were no statistically significant differences in the FF values of all three muscles between the normal side and the disc displacement side ($p > 0.05$) (Table 4).

Within the TMD subgroups, only the masseter muscle in the subgroup without clenching had a statistically significantly lower FF value on the disc displacement side (3.94 ± 1.61) than on the normal side (4.52 ± 2.24) ($p < 0.05$, Cohen's d = -0.502) (Figs 2 and 3).

**Table 3. Fat fraction (%) of the normal group (n = 19).**

|  | Right side | Left side | *p*-value | *Effect size (Cohen's d)* |
|---|---|---|---|---|
| **Masseter muscle** | 2.95 ± 0.96 | 2.92 ± 1.12 | 0.857 | -0.042 |
| **Medial pterygoid muscle** | 2.04 ± 1.18 | 1.83 ± 0.68 | 0.365 | -0.213 |
| **Lateral pterygoid muscle** | 2.88 ± 1.13 | 2.92 ± 1.24 | 0.844 | 0.046 |

The data are presented as mean ± standard deviation. The paired t-test was used.

**Table 4. Fat fraction (%) of the TMD group (n = 46).**

|  | Normal side | Disc displacement side | *p*-value | *Effect size (Cohen's d)* |
|---|---|---|---|---|
| **Masseter muscle** | 3.86 ± 1.96 | 3.68 ± 1.65 | 0.303 | -0.154 |
| **Medial pterygoid muscle** | 2.74 ± 1.22 | 2.48 ± 1.15 | 0.089 | -0.256 |
| **Lateral pterygoid muscle** | 2.60 ± 1.23 | 2.77 ± 1.41 | 0.323 | 0.147 |

The data are presented as mean ± standard deviation. The paired t-test was used.

## Discussion

This study aimed to quantitatively evaluate the FF value of the masticatory muscles using CSE-MRI and determine whether this method could be a new method for muscle evaluation of TMDs. TMDs result from abnormalities in the function of the TMJ due to various factors. TMDs can be caused by muscle abnormalities, while other bone and disc abnormalities can also lead to changes in the muscles. Pain, a common symptom of TMDs, can cause myalgia and myospasm in the masticatory muscles. Abnormal habits such as clenching and bruxism are also related to these changes. It has been reported that children with myofascial TMD pain and bruxism had altered temporal and masseter muscle electromyographic activity at rest and during maximum clenching, compared to children without TMD [29].

Studies have been conducted on muscle thickness using ultrasound [11–13], CT [7, 8], and MRI [9, 10], and research has investigated muscle elasticity through ultrasound elastography [13], but there have been few studies on changes in muscle components using imaging. Muraoka et al. [30] recently reported that MRI texture analysis of the lateral pterygoid muscles may be useful for differentiating osteoarthritis from rheumatoid arthritis of the TMJ. Chen et al. [28] found that after right maxillary molar extraction in rabbits, the FF values of both masseter muscles were higher at 12 weeks than at 0 weeks, suggesting that FF values could be used as indicators of masticatory muscle dysfunction. Our study represents the first research on masticatory muscles in TMD patients using the CSE-MRI technique. Disc displacement and habitual clenching in TMD patients are also expected to affect the muscles, potentially causing changes in muscle composition or thickness. However, noticeable changes in muscle thickness may take a significant amount of time to develop. The muscle fat fraction can be calculated using the CSE-MRI sequence. In a previous study on the condylar head, the FF of the TMD group (60.56%) was statistically significantly lower than that of the normal group (78.30%) [27]. Accuracy and reproducibility were demonstrated for FF measurements of the mandibular condyle [27] and parotid gland [23], and our study of the masticatory muscles also exhibited excellent intra- and inter-observer agreement.

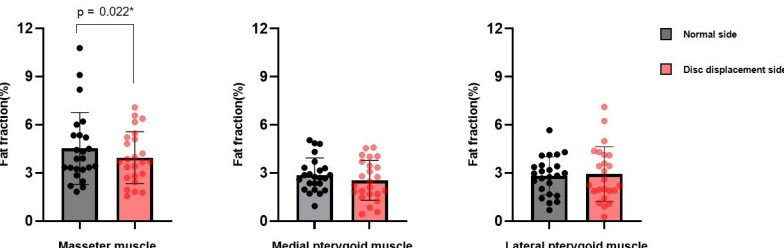

**Fig 2. Fat fraction (%) compared between the normal side and disc displacement side of the TMD group without clenching.** The paired t-test was used, * $p < 0.05$.

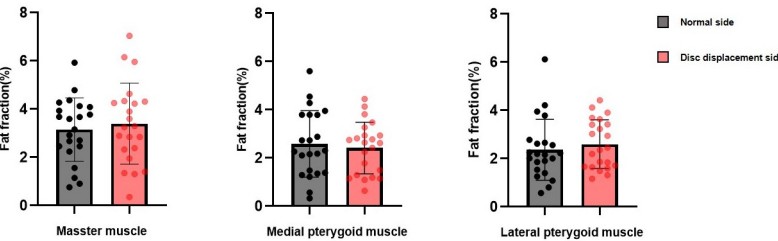

**Fig 3. Fat fraction (%) compared between the normal side and disc displacement side of the TMD group with clenching.** The paired t-test was used.

Since the FF of body parts can vary depending on various factors, such as sex, age, and body mass index (BMI) [22], it may be inaccurate to simply compare the FF values of the masticatory muscles between the normal group and the TMD group. Therefore, we first confirmed whether there was a difference in the FF values of the right and left masticatory muscles, and found no statistical difference in the masseter, medial pterygoid, and lateral pterygoid muscles. The factors affecting masticatory muscles are diverse and complex, and we set up a controllable situation as the first study on FF values in the masticatory muscles. We compared the normal side and the disc displacement side in TMD patients with unilateral disc displacement, but unexpectedly, there was no statistical difference in all three masticatory muscles. This may be due to the small sample size and various factors such as clenching habits, in addition to disc displacement, due to the complexity of TMDs. Therefore, we reanalyzed the FF values according to the presence or absence of clenching habits. In the TMD group without clenching, only the masseter muscle FF values of the disc displacement side (3.94 ± 1.61) were statistically significantly lower than those of the normal side (4.52 ± 2.24). In the TMD group with clenching, there was no statistical difference between the muscle FF values of the disc displaced side and the normal side. It is thought that the FF value of the masseter muscle is helpful in diagnosing TMD in the group with only the effect of disc displacement, without the influence of clenching. In a previous study, the apparent diffusion coefficient (ADC) values on MRI of the masticatory muscles on the painful and non-painful sides were compared in 106 TMD patients with unilateral pain, and the mean ADC values of the muscles on the painful side were significantly greater than those on the non-painful sides ($p < 0.01$) [31]. Yanagisawa et al. [32] reported that an increase in ADC value on the painful side was indicative of edematous changes and indicated an increase in intramuscular water transport, including water diffusion in the extravascular space and microcirculation in the capillary network. In our study, it is thought that the lower FF value on the disc displacement side reflects a relative decrease in fat content due to increased water content resulting from edematous changes caused by pain. Additionally, muscle use restriction due to pain is presumed to have had an effect.

Our current study has some limitations. First, the sample size was small. Future research should include a larger number of participants, taking into account factors such as age, sex, and BMI. Second, FF values were measured using a single slice of muscle; thus, future studies should measure FFs in multiple slices that include whole muscle. Finally, further studies should consider the numerous variables that can affect muscles beyond clenching and disc displacement, such as muscle pain, muscle stiffness, and bite force. It would also be interesting to analyze changes in FF values over the course of muscle treatment in the same person.

In conclusion, we have confirmed that the FF value can be employed as a tool for the quantitative analysis of the masticatory muscles, including the masseter, medial, and lateral pterygoid muscles. There was no significant difference in FF values between the normal side and

displaced side in TMD patients with unilateral disc displacement. However, when comparing only patients without clenching, differences in FF values were found in the masseter muscle. Therefore, we determined that the FF value of the masseter muscle exhibited greater applicability for the clinical diagnosis of patients with TMDs, and further studies are needed.

## Author Contributions

**Conceptualization:** Kug Jin Jeon, Chena Lee, Sang-Sun Han.

**Data curation:** Kug Jin Jeon, Yoon Joo Choi, Chena Lee, Hak-Sun Kim.

**Formal analysis:** Kug Jin Jeon, Yoon Joo Choi.

**Funding acquisition:** Sang-Sun Han.

**Investigation:** Kug Jin Jeon, Yoon Joo Choi, Chena Lee.

**Methodology:** Kug Jin Jeon, Sang-Sun Han.

**Project administration:** Sang-Sun Han.

**Supervision:** Kug Jin Jeon, Sang-Sun Han.

**Validation:** Kug Jin Jeon, Yoon Joo Choi, Chena Lee, Hak-Sun Kim, Sang-Sun Han.

**Visualization:** Kug Jin Jeon.

**Writing – original draft:** Kug Jin Jeon, Sang-Sun Han.

**Writing – review & editing:** Kug Jin Jeon, Sang-Sun Han.

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
