## [Decision Letter · Decision Letter 0]

6 Nov 2023

PONE-D-23-30725Evaluation of masticatory muscles in temporomandibular joint disorder patients using quantitative MRI fat fraction analysis - Could it be a biomarker?PLOS ONE

Dear Dr. Han,

Thank you for submitting your manuscript to PLOS ONE. After careful consideration, we feel that it has merit but does not fully meet PLOS ONE’s publication criteria as it currently stands. Therefore, we invite you to submit a revised version of the manuscript that addresses the points raised during the review process.

We look forward to receiving your revised manuscript.

Kind regards,

Rocco Franco

Academic Editor

PLOS ONE

Journal Requirements:

Yonsei University College of Dentistry (6-2023-0005)

This study was supported by the Yonsei University College of Dentistry (6-2023-0005). 

Yonsei University College of Dentistry (6-2023-0005)

Additional Editor Comments:

Dear Authors,

Please follow the reviewer instruction.

Regards

Reviewers' comments:

Reviewer's Responses to Questions

**Comments to the Author**

1. Is the manuscript technically sound, and do the data support the conclusions?

Reviewer #1: Yes

Reviewer #2: Yes

Reviewer #3: Yes

2. Has the statistical analysis been performed appropriately and rigorously? 

Reviewer #1: No

Reviewer #2: Yes

Reviewer #3: Yes

3. Have the authors made all data underlying the findings in their manuscript fully available?

Reviewer #1: Yes

Reviewer #2: Yes

Reviewer #3: Yes

4. Is the manuscript presented in an intelligible fashion and written in standard English?

Reviewer #1: Yes

Reviewer #2: Yes

Reviewer #3: Yes

5. Review Comments to the Author

Reviewer #1: Review of a study titled: Evaluation of masticatory muscles in temporomandibular joint disorder

patients using quantitative MRI fat fraction analysis - Could it be a biomarker?

I think the article is interesting. However, it needs to improve the statistical analysis, add missing data, compare between groups (size and age), calculate sample size, add effect size. Also, please expand the introduction according to my comments.

The title and abstract in my opinion is correct.

ll31-33 –‘’ The muscle is evaluated throughpalpation, for which it is important to apply a uniform force using the finger. Nevertheless, this methodis subjective and may vary depending on the evaluator’’ - Standardization of the palpation examination is described in the RDC/TMD and DC/TMD protocols. Describe the examination protocol in the introduction. Refer to these research methods in the introduction. DOI: 10.11607/jop.1151

ll35 – ‘’ few studies have measured muscle thickness using CT [5] and MRI [6]’’ - The authors write about studies in the plural and refer to one example. This is a mistake, add additionally one study each for CT (for example. DOI: 10.7417/CT.2019.2147 ) and for MRI (For example DOI: 10.3390/jcm12124166 ).

ll39 - Be sure to add citations to support your sentences. In this form, it is only the authors' opinion.

Introduction - Add the latest epiemiological data on TMD. DOI: 10.1007/s00784-020-03710-w Add a hypothesis and an alternative hypothesis at the end.

In addition, when it comes to using the abbreviation TMD, I suggest the abbreviation TMDs.

Statistical analysis - Add sample size calculation information. In addition, add effect size to statistically significant results as recommended: doi: 10.4300/JGME-D-12-00156.1

Statistical analysis and Results - add comparisons of the number of women and men, age between the study groups.

ll153 – ‘’ Studies’’ - According to my earlier comment. Add again after the study for CT, MRI and ultrasound.

Reviewer #2: Dear Authors,

Thank you very much for the opportunity to review on such an important topic. Temporomandibular Joint Disorder (TMD) is a complex condition that affects the temporomandibular joint and associated muscles, causing pain and discomfort for many individuals.

The study, which included 65 patients who underwent 3T MRI scans, is a significant step forward in understanding and diagnosing TMD. Patients were categorized into a normal group and a TMD group, with the latter further divided based on the presence or absence of clenching behavior. The researchers meticulously evaluated the right and left FF values of the masseter, medial, and lateral pterygoid muscles using CSE-MRI and recorded their findings.

One of the notable strengths of this study was the excellent intra- and inter-observer agreement in measuring FF using CSE-MRI, with an Intraclass Correlation Coefficient (ICC) of over 0.889 for both, highlighting the method's reproducibility and reliability.

In the TMD group without clenching, a statistically significant difference was observed in the masseter muscle's FF values, with a lower FF value on the side affected by disc displacement compared to the unaffected side. This finding suggests that CSE-MRI, with its ability to reproducibly quantify muscle FF values, holds great promise as a biomarker for objectively assessing muscle conditions in TMD patients, particularly within the masseter muscle.

The study's results raise the possibility of using CSE-MRI as a valuable tool in diagnosing and monitoring TMD patients. You could also refere in your discussion and enrich the work and make it even better for a wider audience to DOI: 10.3390/jcm11051323.

Reviewer #3: The present study investigated human masticatory muscles using fat fraction (FF) values; the authors aimed to confirm the usefulness of FF values in evaluating the masticatory muscles of temporomandibular disorder (TMD) patients.

The idea of the manuscript is original and has clinical relevance for the field of dentistry.

a) Introduction: The introduction is objective and the literature presented was well referenced.

b) Materials and Methods: The ethical authorization was presented in the text. The experiment design was well described and well conducted with valid methodology, but some issues deserves be pointed.

1 - The authors didn’t present sample size calculation. In this way in order to ensure that the convenience sample of this study was representative, we suggest that a sample calculation (post-hoc) be carried out.

2 - I think it is important for the authors to justify why they only used MRI in the period from August 2022 to March 2023.

3 - The authors reported that the presence of clenching (or bruxism) that affects the masticatory muscles was clinically assessed. Clenching and bruxism are different. The polysomnography exam is the gold standard for diagnosing bruxism. If the individuals were not submitted to this exam the authors cannot confirm the presence of bruxism. Please verify this statement.

4 - Regarding statistical analysis, the authors used the Wicoxon test and paired t-test. Which test was used to check the sample distribution? This information must be included in this section.

c) Results and Discussion

5 - Considering that a non-parametric test was used for the normal group, the data in Table 3 must be represented appropriately (medians and quartiles).

6 - The authors should take into account the gender factor in the TMD group and carry out a statistical analysis considering only women. This data should be included in the results and discussed.

7 – The captions for figures 2 and 3 must be corrected in relation to the statistical test.

6. PLOS authors have the option to publish the peer review history of their article (what does this mean?). If published, this will include your full peer review and any attached files.

Reviewer #1: No

Reviewer #2: No

Reviewer #3: No

---

## [Author Response · Author response to Decision Letter 0]

8 Dec 2023

Reviewer 1:

Comment 1: Review of a study titled: Evaluation of masticatory muscles in temporomandibular joint disorder patients using quantitative MRI fat fraction analysis - Could it be a biomarker?

I think the article is interesting. However, it needs to improve the statistical analysis, add missing data, compare between groups (size and age), calculate sample size, add effect size. Also, please expand the introduction according to my comments.

The title and abstract in my opinion is correct.

Reply 1: We revised the manuscript based on your advice. In particular, we made revisions in response to comments 2-8 below.

Comment 2: ll31-33 –‘’ The muscle is evaluated through palpation, for which it is important to apply a uniform force using the finger. Nevertheless, this methods subjective and may vary depending on the evaluator’’ - Standardization of the palpation examination is described in the RDC/TMD and DC/TMD protocols. Describe the examination protocol in the introduction. Refer to these research methods in the introduction. DOI: 10.11607/jop.1151

Reply 2: In accordance with your advice, we revised the text. (See page 3, lines 33-36)

Changes in the text: We highlighted the changed parts.

“Muscles are evaluated by palpation with 1 kg of pressure for 2 seconds, and if pain is present, the pressure is prolonged for 5 seconds to allow more time to elicit spreading or referred pain [6]. However, these methods are not objective and cannot be quantified, and assessing subtle changes in muscles can be difficult.”

Reference) 6. Schiffman E, Ohrbach R, Truelove E, Look J, Anderson G, Goulet J, et al. International RDC/TMD consortium network, international association for dental research; orofacial pain special interest group, international association for the study of pain. J Oral Facial Pain Headache. 2014;28(1):6-27. (DOI: 10.11607/jop.1151)

Comment 3: ll35 – ‘’ few studies have measured muscle thickness using CT [5] and MRI [6]’’ - The authors write about studies in the plural and refer to one example. This is a mistake, add additionally one study each for CT (for example. DOI: 10.7417/CT.2019.2147) and for MRI (For example DOI: 10.3390/jcm12124166 ).

Reply 3: In accordance with your advice, we added studies. (See page 3, lines 36-39)

Changes in the text: We highlighted the changed parts.

“To date, there is no quantitative evaluation method for muscles, although a few studies have measured muscle thickness using CT [7, 8] and MRI [9, 10]. Some research has assessed the thickness and elasticity of masseter muscles using ultrasound [11-13], ~”

Reference) 

8. Yang SM, Wu HW, Lin YH, Lai TJ, Lin MT. Temporalis and masseter muscle thickness as predictors of post-stroke dysphagia after endovascular thrombectomy. Eur J Radiol. 2023;165:110939. 

10. Zieliński G, Matysik-Woźniak A, Pankowska A, Pietura R, Rejdak R, Jonak K. High Myopia and Thickness of Extraocular and Masticatory Muscles-7T MRI, Preliminary Study. J Clin Med. 2023;12(12):4166. (DOI: 10.3390/jcm12124166)

11. Impellizzeri A, Serritella E, Putrino A, Vizzielli G, Polimeni A, Galluccio G. Assessment of Masticatory and Cervical Muscles' Thickness by Ultrasonography in Patients with Facial Asymmetry. Clin Ter. 2019;170(4):e272-7. (DOI: 10.7417/CT.2019.2147)

Comment 4: ll39 - Be sure to add citations to support your sentences. In this form, it is only the authors' opinion.

Reply 4: We added citations. (See page 3, lines 40-42)

Changes in the text: We highlighted the changed parts.

“Furthermore, ultrasound signals cannot pass through bones and can only evaluate the superficially located masseter muscle and temporalis muscles, and it is unable to assess the deep-seated pterygoid muscles [15].”

Reference) 15. Pereira LJ, Gavião MB, Bonjardim LR, Castelo PM, van der Bilt A. Muscle thickness, bite force, and craniofacial dimensions in adolescents with signs and symptoms of temporomandibular dysfunction. Eur J Orthod. 2007;29(1):72-8.

Comment 5: Introduction - Add the latest epidemiological data on TMD. DOI: 10.1007/s00784-020-03710-w Add a hypothesis and an alternative hypothesis at the end.

Reply 5: In accordance with your advice, we added this reference and the latest epidemiological data. (See page 3, lines 26-28) 

We also added a hypothesis and an alternative hypothesis at the end. (See page 4, lines 50-52 & page 11, lines 212-214)

Changes in the text: We highlighted the changed parts.

 “The prevalence of temporomandibular joint disorders (TMDs) is gradually increasing and the age of onset is decreasing, with TMDs present in approximately 31% of adults and older adults and 11% of children and adolescents [1].”

“We hypothesized that in the masticatory muscles, the FF value on the disc displacement side would be different from the FF value of the normal side, but that in subjects with clenching, there would be no difference between both sides.”

“There was no significant difference in FF values between the normal side and displaced side in TMD patients with unilateral disc displacement. However, when comparing only patients without clenching, differences in FF values were found in the masseter muscle.”

Reference) 1. Valesan LF, Da-Cas CD, Réus JC, Denardin ACS, Garanhani RR, Bonotto D, et al. Prevalence of temporomandibular joint disorders: a systematic review and meta-analysis. Clin Oral Investig. 2021;25(2):441-53. (DOI: 10.1007/s00784-020-03710-w)

Comment 6: In addition, when it comes to using the abbreviation TMD, I suggest the abbreviation TMDs.

Reply 6: In accordance with your advice, we used the abbreviation TMDs.

Changes in the text: We highlighted the changed parts.

Comment 7: Statistical analysis - Add sample size calculation information. In addition, add effect size to statistically significant results as recommended: doi: 10.4300/JGME-D-12-00156.1

Reply 7: We added sample size calculation information. (See page 4, lines 66-67)

Changes in the text: We highlighted the changed parts.

“Using the G*Power program, the sample size was calculated based on previous study (power=0.9, α=0.05) [23].”

Comment 8: Statistical analysis and Results - add comparisons of the number of women and men, age between the study groups.

Reply 8: We compared the number of women and men, as well as the age between the study groups, in Table 1. (See page 5, lines 80-82)

In our study, we compared the left and right sides of the same person, regardless of gender and age. Due to the small number of samples, we did not compare masticatory muscles of other people by gender or age, and this was described as a limitation. (See page 10, lines 203-204)

Changes in the text: We highlighted the changed parts.

Table 1. Clinical characteristics of the normal group and the TMD group with unilateral disc displacement 

“Our current study has some limitations. First, the sample size was small. Future research should include a larger number of participants, taking into account factors such as age, sex, and BMI.”

Comment 9: ll153 – ‘’ Studies’’ - According to my earlier comment. Add again after the study for CT, MRI and ultrasound.

Reply 9: We added studies for CT, MRI and ultrasound. (See page 9, lines 162)

Changes in the text: We highlighted the changed parts.

“Studies have been conducted on muscle thickness using ultrasound [11-13], CT [7, 8], and MRI [9, 10], ~”

Reviewer 2:

Comments to Author

Thank you very much for the opportunity to review on such an important topic. Temporomandibular Joint Disorder (TMD) is a complex condition that affects the temporomandibular joint and associated muscles, causing pain and discomfort for many individuals.

The study, which included 65 patients who underwent 3T MRI scans, is a significant step forward in understanding and diagnosing TMD. Patients were categorized into a normal group and a TMD group, with the latter further divided based on the presence or absence of clenching behavior. The researchers meticulously evaluated the right and left FF values of the masseter, medial, and lateral pterygoid muscles using CSE-MRI and recorded their findings.

One of the notable strengths of this study was the excellent intra- and inter-observer agreement in measuring FF using CSE-MRI, with an Intraclass Correlation Coefficient (ICC) of over 0.889 for both, highlighting the method's reproducibility and reliability.

In the TMD group without clenching, a statistically significant difference was observed in the masseter muscle's FF values, with a lower FF value on the side affected by disc displacement compared to the unaffected side. This finding suggests that CSE-MRI, with its ability to reproducibly quantify muscle FF values, holds great promise as a biomarker for objectively assessing muscle conditions in TMD patients, particularly within the masseter muscle.

The study's results raise the possibility of using CSE-MRI as a valuable tool in diagnosing and monitoring TMD patients. 

Comment 1: You could also refer in your discussion and enrich the work and make it even better for a wider audience to DOI: 10.3390/jcm11051323.

Reply 1: We added a reference to DOI: 10.3390/jcm11051323 and included it in our discussion. (See page 9, lines 158-161)

Changes in the text: We highlighted the changed parts.

“It has been reported that children with myofascial TMD pain and bruxism had altered temporal and masseter muscle electromyographic activity at rest and during maximum clenching, compared to children without TMD [29].” 

Reference) 29. Szyszka-Sommerfeld L, Sycińska-Dziarnowska M, Budzyńska A, Woźniak K. Accuracy of Surface Electromyography in the Diagnosis of Pain-Related Temporomandibular Disorders in Children with Awake Bruxism. J Clin Med. 2022;11(5):1323. (DOI: 10.3390/jcm11051323)

Reviewer 3:

The present study investigated human masticatory muscles using fat fraction (FF) values; the authors aimed to confirm the usefulness of FF values in evaluating the masticatory muscles of temporomandibular disorder (TMD) patients.

The idea of the manuscript is original and has clinical relevance for the field of dentistry.

a) Introduction: The introduction is objective and the literature presented was well referenced.

b) Materials and Methods: The ethical authorization was presented in the text. The experiment design was well described and well conducted with valid methodology, but some issues deserves be pointed.

Comment 1: The authors didn’t present sample size calculation. In this way in order to ensure that the convenience sample of this study was representative, we suggest that a sample calculation (post-hoc) be carried out.

Reply 1: We added information on the sample size calculation. (See page 4, lines 66-67)

Changes in the text: We highlighted the changed parts. 

“Using the G*Power program, the sample size was calculated based on previous study (power=0.9, α=0.05) [23].”

Comment 2: I think it is important for the authors to justify why they only used MRI in the period from August 2022 to March 2023.

Reply 2: We set a period with a sample number that met the pre-calculated sample size. We conducted a pilot study to use FF in masticatory muscle assessment of TMDs to confirm its feasibility with a minimal sample size, and this study confirmed its feasibility for use as a biomarker. Further research will be conducted using a large sample, and we described this point as a limitation. (See page 10, lines 203-204)

Changes in the text: We highlighted the changed parts.

“Our current study has some limitations. First, the sample size was small. Future research should include a larger number of participants, taking into account factors such as age, sex, and BMI.”

Comment 3: The authors reported that the presence of clenching (or bruxism) that affects the masticatory muscles was clinically assessed. Clenching and bruxism are different. The polysomnography exam is the gold standard for diagnosing bruxism. If the individuals were not submitted to this exam the authors cannot confirm the presence of bruxism. Please verify this statement.

Reply 3: We appreciate your advice. Although the polysomnography exam is the gold standard for diagnosing bruxism, it cannot be performed on all patients, so we have specifically explained the clinical evaluation and added references. We will proceed according to your opinion in further research. (See page 5, lines 74-77)

Changes in the text: We highlighted the changed parts.

“In the TMD group, the presence of clenching (or bruxism) that affected the masticatory muscles was diagnosed through self-report of clenching and a clinical examination to evaluate clenching signs, such as abnormal tooth wear and teeth impressions in the buccal region [29].”

Comment 4: Regarding statistical analysis, the authors used the Wicoxon test and paired t-test. Which test was used to check the sample distribution? This information must be included in this section.

Reply 4: As your advice, we confirmed the normality of data distribution in all groups using the Kolmogorov-Smirnov test and statistically re-analyzed the Wilcoxon test using the paired t-test in the normal group, subgroup without clenching, and subgroup with clenching. The entire manuscript has been revised accordingly. (See 6, lines 106-108)

Changes in the text: We highlighted the changed parts.

“The normality of data distribution was confirmed using the Kolmogorov-Smirnov test in all groups, including the normal group, TMD group, the subgroup without clenching, and the subgroup with clenching.”

c) Results and Discussion:

Comment 5: Considering that a non-parametric test was used for the normal group, the data in Table 3 must be represented appropriately (medians and quartiles).

Reply 5: We confirmed the normality of data distribution in all groups using the Kolmogorov-Smirnov test and statistically re-analyzed the Wilcoxon test using the paired t-test in the normal group (See 6, lines 106-108)

Changes in the text: We highlighted the changed parts.

“The normality of data distribution was confirmed using the Kolmogorov-Smirnov test in all groups, including the normal group, TMD group, the subgroup without clenching, and the subgroup with clenching.”

Comment 6: The authors should take into account the gender factor in the TMD group and carry out a statistical analysis considering only women. This data should be included in the results and discussed.

Reply 6: Unfortunately, this study was a pilot study and did not consider the gender factor. In our study, we compared the left and right sides of the same person, regardless of gender and age. Due to the small number of samples, we did not compare masticatory muscles of other people by gender or age. This was described as limitation and will be considered in future research. (See page 9, lines 178-181 & page 10, lines 203-204)

Changes in the text: We highlighted the changed parts.

“Since the FF of body parts can vary depending on various factors, such as sex, age, and body mass index (BMI) [22], it may be inaccurate to simply compare the FF values of the masticatory muscles between the normal group and the TMD group. Therefore, we first confirmed whether there was a difference in the FF values of the right and left masticatory muscles, ~”

“Our current study has some limitations. First, the sample size was small. Future research should include a larger number of participants, taking into account factors such as age, sex, and BMI.”

Comment 7: The captions for figures 2 and 3 must be corrected in relation to the statistical test.

Reply 7: As your advice, we confirmed the normality of the data distribution in all groups using the Kolmogorov-Smirnov test and re-statistically analyzed the Wilcoxon test using the paired t-test in the normal group, subgroup without clenching, and subgroup with clenching. The entire manuscript has been revised accordingly. We revised Table 3, Figure 2, and Figure 3 and revised the p-values based on the paired t-test results. (See page 7, lines 133-134 & page 8, lines 146-150)

Changes in the text: We highlighted the changed parts.

Table 3. Fat fraction (%) of the normal group (n = 19)

The data are presented as mean ± standard deviation. The paired t-test was used.

Fig 2. Fat fraction (%) compared between the normal side and disc displacement side of the TMD group without clenching. The paired t-test was used, * p < 0.05.

Fig 3. Fat fraction (%) compared between the normal side and disc displacement side of the TMD group with clenching. The paired t-test was used.

---

## [Decision Letter · Decision Letter 1]

12 Dec 2023

PONE-D-23-30725R1Evaluation of masticatory muscles in temporomandibular joint disorder patients using quantitative MRI fat fraction analysis - Could it be a biomarker?PLOS ONE

Dear Dr. Han,

Thank you for submitting your manuscript to PLOS ONE. After careful consideration, we feel that it has merit but does not fully meet PLOS ONE’s publication criteria as it currently stands. Therefore, we invite you to submit a revised version of the manuscript that addresses the points raised during the review process.

Dear Authors,

Please revise according Reviewer's 1 recommendation

We look forward to receiving your revised manuscript.

Kind regards,

Rocco Franco

Academic Editor

PLOS ONE

Journal Requirements:

Additional Editor Comments:

Dear Authors,

Please change this manuscript according Reviewer 1.

Regards

Reviewers' comments:

Reviewer's Responses to Questions

**Comments to the Author**

1. If the authors have adequately addressed your comments raised in a previous round of review and you feel that this manuscript is now acceptable for publication, you may indicate that here to bypass the “Comments to the Author” section, enter your conflict of interest statement in the “Confidential to Editor” section, and submit your "Accept" recommendation.

Reviewer #1: (No Response)

Reviewer #2: All comments have been addressed

2. Is the manuscript technically sound, and do the data support the conclusions?

Reviewer #1: Yes

Reviewer #2: Yes

3. Has the statistical analysis been performed appropriately and rigorously? 

Reviewer #1: No

Reviewer #2: Yes

4. Have the authors made all data underlying the findings in their manuscript fully available?

Reviewer #1: Yes

Reviewer #2: Yes

5. Is the manuscript presented in an intelligible fashion and written in standard English?

Reviewer #1: Yes

Reviewer #2: Yes

6. Review Comments to the Author

Reviewer #1: Thank you for your reply. For the most part, the authors responded correctly to my comments.

Except for one point. The authors did not add the effect size to the results. The effect size is not the same as the effect size given to the sample size calculation. Please provide the effect size for each outcome. doi: 10.4300/JGME-D-12-00156.1

Reviewer #2: I believe that the findings presented in the research will make a valuable contribution to the Journal and enhance the understanding this subject.Thank you for your step-by-step responses to all the reviewers' comments.

7. PLOS authors have the option to publish the peer review history of their article (what does this mean?). If published, this will include your full peer review and any attached files.

Reviewer #1: No

Reviewer #2: No

---

## [Author Response · Author response to Decision Letter 1]

17 Dec 2023

Journal Requirements:

Reply 1: We reviewed the reference list to ensure that it is complete and correct. We did not cite the retracted paper and modified it to fit the form.

Changes in the text: We highlighted the changed parts.

6. Schiffman E, Ohrbach R, Truelove E, Look J, Anderson G, Goulet JP, et al. Diagnostic Criteria for Temporomandibular Disorders (DC/TMD) for Clinical and Research Applications: recommendations of the International RDC/TMD Consortium Network* and Orofacial Pain Special Interest Group†. J Oral Facial Pain Headache. 2014;28(1):6-27.

Reviewer 1:

Comment 1: Thank you for your reply. For the most part, the authors responded correctly to my comments.

Except for one point. The authors did not add the effect size to the results. The effect size is not the same as the effect size given to the sample size calculation. Please provide the effect size for each outcome. doi: 10.4300/JGME-D-12-00156.1

Reply 1: We added the effect size to the results based on your advice. (See page 8, lines 138, 144 and 147-149).

Changes in the text: We highlighted the changed parts.

Table 3. Fat fraction (%) of the normal group (n = 19)

Table 4. Fat fraction (%) of the TMD group (n = 46) 

“Within the TMD subgroups, only the masseter muscle in the subgroup without clenching had a statistically significantly lower FF value on the disc displacement side (3.94 ± 1.61) than on the normal side (4.52 ± 2.24) (p < 0.05, Cohen’s d = -0.502) (Fig. 2, Fig. 3).”

---

## [Decision Letter · Decision Letter 2]

19 Dec 2023

Evaluation of masticatory muscles in temporomandibular joint disorder patients using quantitative MRI fat fraction analysis - Could it be a biomarker?

PONE-D-23-30725R2

Dear Dr. Han,

We’re pleased to inform you that your manuscript has been judged scientifically suitable for publication and will be formally accepted for publication once it meets all outstanding technical requirements.

Kind regards,

Rocco Franco

Academic Editor

PLOS ONE

Additional Editor Comments (optional):

Reviewers' comments:

Reviewer's Responses to Questions

**Comments to the Author**

1. If the authors have adequately addressed your comments raised in a previous round of review and you feel that this manuscript is now acceptable for publication, you may indicate that here to bypass the “Comments to the Author” section, enter your conflict of interest statement in the “Confidential to Editor” section, and submit your "Accept" recommendation.

Reviewer #1: All comments have been addressed

2. Is the manuscript technically sound, and do the data support the conclusions?

Reviewer #1: (No Response)

3. Has the statistical analysis been performed appropriately and rigorously? 

Reviewer #1: (No Response)

4. Have the authors made all data underlying the findings in their manuscript fully available?

Reviewer #1: (No Response)

5. Is the manuscript presented in an intelligible fashion and written in standard English?

Reviewer #1: (No Response)

6. Review Comments to the Author

Reviewer #1: (No Response)

7. PLOS authors have the option to publish the peer review history of their article (what does this mean?). If published, this will include your full peer review and any attached files.

Reviewer #1: No

---

## [Editor Report · Acceptance letter]

10 Jan 2024

PONE-D-23-30725R2 

PLOS ONE

Dear Dr. Han, 

I'm pleased to inform you that your manuscript has been deemed suitable for publication in PLOS ONE. Congratulations! Your manuscript is now being handed over to our production team.

Kind regards, 

on behalf of

Dr. Rocco Franco 

Academic Editor

PLOS ONE